Genome-wide identification and expression analysis of the VQ gene family in Cucurbita pepo L.

Xu Ke
Wang Ping wangping@imau.edu.cn
College of Horticulture and Plant Protection, Inner Mongolia Agricultural University , Huhehaote, Inner Mongolia , China
Uversky Vladimir
Electronic publication date: 2022 Jan 21
Publication date: 2022
Volume: 10
Electronic Location ID: e12827
Received 2021 Sep 6; Accepted 2022 Jan 3
Copyright: © 2022 Xu and Wang
Copyright year: 2022
Copyright holder: Xu and Wang
License: This is an open access article distributed under the terms of the Creative Commons Attribution License, which permits unrestricted use, distribution, reproduction and adaptation in any medium and for any purpose provided that it is properly attributed. For attribution, the original author(s), title, publication source (PeerJ) and either DOI or URL of the article must be cited.
License URL: https://creativecommons.org/licenses/by/4.0/

Keywords: VQ genes family, Expression pattern, Powdery mildew, Cucurbita pepo L.

Funding: Inner Mongolia Science and Technology 2021GG0148, 20120212 Inner Mongolia Agricultural University Animal and Plant Varieties Breeding (Cultivation) YZGC2017018 Inner Mongolia Natural Science Foundation 2021MS03050 This work was supported by the Inner Mongolia science and technology plan project (No. 2021GG0148, 20120212); and the Inner Mongolia Agricultural University Animal and plant varieties breeding (cultivation) special project (No. YZGC2017018); the Inner Mongolia Natural Science Foundation Project (No. 2021MS03050); The funders had no role in study design, data collection and analysis, decision to publish, or preparation of the manuscript.

==============================
VQ protein is a plant specific protein, which plays an important role in plant growth and development and biological and abiotic stress response. This study aimed to systematically analyze for the first time the VQ of Cucurbita pepo and understand their expression patterns in response to different stimuli. Herein, 44 VQ genes were identified, which were divided into eight groups (I–VIII) based on phylogenetic analysis. Two genes (CpVQ1 and CpVQ2) could not be located on the chromosome, whereas the remaining CpVQ genes were randomly distributed on the chromosomes, except for chromosomes 15 and 18. Noteworthy, the main event driving the expansion of the VQ gene family was chromosome fragment duplication. Based on qRT-PCR analysis, VQ genes are expressed in different tissues, and VQ genes are differentially regulated under a variety of abiotic stresses and powdery mildew stress, indicating that they play an important role in plant stress response and other aspects. This report presents the first systematic analysis of VQ genes from C. pepo and provides a solid foundation for further research of the specific functions of VQ proteins.

Introduction

VQ protein is a highly conserved transcription factor that is widely expressed in biology, including several monocotyledonous and dicotyledonous plants (Jing & Lin, 2015; Song et al., 2016; Ding et al., 2019). This protein owns its name to its VQ motifs, which allows it to interact with WRKY transcription factors to regulate plant growth and development (Lai et al., 2011). For example, VQ10 interacts with WRKY25 and WRKY33, which when simultaneously overexpressed can impair plant growth (Cheng et al., 2012). In Arabidopsis, AtWRKY2/34 and AtVQ20 form a complex that negatively regulates the expression of MYB transcription factors to regulate the development and function of pollen (Lei et al., 2017; Lei, Ma & Yu, 2018). The ATVQ8 gene affects leaf development, making the leaves appear yellow-green throughout the growth period, causing plant malnutrition and affecting the normal growth of Arabidopsis plants. Plants overexpressing VQ17, VQ18, and VQ22 have slow growth and yellow leaves, and the leaf area becomes smaller (Cheng et al., 2012). Moreover, IKU1 encodes a protein containing a VQ motif that is involved in the regulation of endosperm development, which in turn affects seed size (Wang et al., 2010a).

VQ plays an important protective role on plants, allowing them to fight pathogenic bacteria and other biological and abiotic stresses (Jiang, Sevugan & Ramachandran, 2018). For example, OsVQ13 is highly expressed at 6 h after bacterial blight infection; thus, it may be involved in the resistance pathway of Oryza sativa to bacterial blight (Kim et al., 2013a). Some PbrVQ genes are differentially expressed under gibberellin and black spot stress (Cao et al., 2018). Furthermore, transgenic Arabidopsis overexpressing AtVQ15 is highly sensitive to salt stress during seed germination and seedling growth, whereas mutants of this gene show strong salt tolerance (Perruc et al., 2004). In response to harsh environments, plants have evolved several complex mechanisms to deal with external stimuli (Fujita et al., 2006). This response is caused by the interaction of transcription factors and cofactors to coordinate the transcription mechanisms of the plant in response to the surrounding environment (Wray et al., 2003). For example, AtVQ9 acts antagonistically with AtWRKY8 to mediate responses to salt stress (Hu et al., 2013). Studies have shown that AtVQ10 and WRKY8 form a complex in the nucleus and positively regulate the resistance of Arabidopsis to Botrytis cinerea (Chen et al., 2018). The banana fruit MaWRKY26 transcription factor physically interacts with the VQ motif-containing protein MaVQ5, which leads to attenuated MaWRKY26-induced transactivation of jasmonic acid biosynthetic genes that are associated with various stress responses (Ye et al., 2016).

C. pepo belongs to the Cucurbita genus of the Cucurbitaceae family. Because it contains several nutrients, it has high edible and medicinal value, and is widely cultivated worldwide. During the cultivation process, C. pepo is susceptible to various biological and abiotic stresses that will affect the yield and quality of the fruit. Therefore, identification of resistance genes has great significance for improving the yield and quality of C. pepo through molecular breeding. Currently, 34, 39, 75, and 61 VQ genes have been identified in Arabidopsis, O. sativa, Glycine max and Maize, respectively (Cheng et al., 2012; Kim et al., 2013a; Wang et al., 2019; Song et al., 2016). However, there is no detailed description of the VQ genes of C. pepo. In this study, we investigated the VQ genes in the C. pepo genome, and analyzed their phylogeny, gene structure, chromosomal location, and collinearity for a comprehensive analysis. In addition, we also analyzed the expression levels of CpVQs under different abiotic stresses and powdery mildew stress. This study is expected to provide basic information for the identification and classification of CpVQ genes. Further experimental analysis can allow us to understand the functions of CpVQs involved in plant stress response.

Materials and Methods

Identification of VQ genes in C. pepo

In present study, two methods were used to identify complete VQ members in C. pepo. Firstly, The Hidden Markov Model (HMM) profiles of the VQ motif (PF05678) was downloaded from PFAM protein family database (http://pfam.xfam.org/) (Punta et al., 2004). Then the model as a probe to perform a BLASTp against the C. pepo genome database using HMMER 3.0. The cut-off E-value was set as 0.1. Secondly, The known VQ motif-containing members of A. thaliana were obtained from The Arabidopsis Information Resource (TAIR) and database (http://arabidopsis.org) were used as queries to conduct a local BLASTp against the protein database of C. pepo. The targeting genes with similarity of E-value less than 1e–20 were retained for the further analysis. We set e as two values of 0.1 and 1e–20 for filtering. We set E-value to 0.1 for the initial filtering, and then set E-value to 1e–20 for in-depth filtering. After merging the two results, the candidate VQ genes were evaluated using the online program PFAM (http://pfam.xfam.org/search), Conserved Domains DataBase Search (https://www.ncbi.nlm.nih.gov/Structure/cdd/wrpsb.cgi) and SMART tool (http://smart.embl-heidelberg.de/) to confirm the presence of VQ motif. The biophysical properties of VQ members, such as, peptide length, molecular weight (MW), isoelectric point (pI) were predicted using the online ExPasy program (http://www.expasy.org/tools/) (Wilkins et al., 1999).

Multiple sequence alignment and phylogenetic analysis

In order to investigate the evolutionary relationships and classification of VQ members in C. pepo, we firstly conducted a multiple sequence alignment based on the full-length amino acid sequence of all CpVQs using the ClustalW tool with auto strategy parameters. According to the alignment results, we constructed a phylogenetic tree using the neighbor-joining (NJ) method with MEGAX software (Tamura et al., 2011). Branch support for the tree topology was estimated by using a bootstrap analysis with 1,000 replicates. The phylogenetic tree was illustrated using the online ITOL (http://itol.embl.de/help.cgi) online tool.

Gene structure analysis and conserved motif detection

The online program Gene Structure Display Server (GSDS) (http://gsds.cbi.pku.edu.cn/) was used to draw a diagrammatic sketch of the intron-exon structure by comparing their coding sequence (CDSs) with their corresponding genomic DNA sequences. The conserved motifs of CpVQs were detected by the online tool MEME (http://meme.ebi.edu.au/). The parameters were set as follows: zero or one occurrence per sequence; maximum number of motifs: 20; and other optional parameters was set default (Bailey et al., 2015).

Chromosome locations, gene duplication and collinearity analysis

For the further investigation of evolution in CpVQs, we firstly constructed the distribution map of VQ members in C. pepo. The physical positions of all VQ genes were retrieved from the GFF3 annotation file using a local script and the schematic diagram of chromosomal location was visualized by TBTOOLS (Chen et al., 2020a). For the identification of tandem duplication events and segmental duplication events in CpVQ family, all CpVQs amino acid sequences were aligned using BLASTp, with an e-value of 1e–10. As previous research described, two or more genes located on the same chromosome were arranged in a 200 kb distance and shared more than 70% identity as analyzed with BLASTp can be defined as tandem duplication events. The segmental duplication events were identified by using the Multiple collinear scanning toolkits (MCScanX) with default parameters (Wang et al., 2012a). The Circos program was used to draw collinearity maps to exhibit duplicated gene pairs between CpVQ genes as well as the synteny blocks of the orthologous VQ genes between C. pepo and Arabidopsis thaliana, Cucumis melo, Oryza sativa, Zea mays.

Calculating Ka and Ks

The Ka and Ks were calculated to assess the selection history and divergence time of gene families. The values of synonymous (Ks) and nonsynonymous (Ka) substitutions of duplicated VQ genes were estimated by using the KaKs_Calculator 2.0 with the NG method (Wang et al., 2010b). The divergence time (T) was calculated using the formula T = Ks/(2 × 6.1 × 10−9) × 10−6 million years ago (MYA) (Kim et al., 2013a, 2013b).

VQ gene expression analysis of C. pepo

The transcriptome expression data of VQ genes under Podosphaera xanthii stress was available from our laboratory. The transcript abundance was represented by fragments per kilobase of exon per million mapped reads (FPKM) values which were calculated based on RNA-Seq reads. The heatmaps showing expression profiles were generated using log10-transformed FPKM values. The results were presented as heatmaps using TBTOOLS software (Chen et al., 2020a).

Plant materials and treatments

Cucurbita pepo (Self-delivered F2) was used in present study. Seeds were planted in a 1:1 (w/w) mixture of soil and sand, cultured in an artificial climatic chamber kept at 30/22 °C with a 18/6 h photoperiod (day/night). Seedlings that germinated after 8 weeks were subjected to different stress conditions: 200 mM NaCl solution, 20% PEG6000 (drought), 4 °C (cold), immersing (waterlogging) and cultured for a total of 24 h, leaves were collected after 0, 12 and 24 h. In order to analyze the expression of VQ genes in different tissues, we collected plant roots, stems, and leaves for RNA preparation. All of the samples were immediately frozen in liquid nitrogen and stored at −80 °C for subsequent total RNA extraction. All samples were tested with three technical replicates and three independent biological replicates.

RNA extraction and quantitative real-time PCR (qRT-PCR)

Total RNA was extracted from C. peop using RNAsimple Total RNA Kit (TIANGEN BIOTECH, Beijing, China) according to the manufacturer’s instructions. First-strand cDNA synthesis was accomplished using TransScript One-Step gDNA Removal and cDNA Synthesis SuperMix (Transgen Biotech, Beijing, China). Quantitative Real-time PCR (qRT-PCR) was performed using TB Green Premix Ex Taq II (TliRNaseH Plus) (RR420Q TaKaRa Biotechnology, Beijing, China) on an FTC-3000P system (Funglyn Biotech, Toronto, Canada) with the primers listed in Table S1. The reaction procedure was completed under the following program: 30 s of pre-denaturation at 95 °C, 40 cycles of 5 s at 95 °C, and 30 s at 60 °C, and 4 °C to finish. All samples were tested with three technical replicates and three independent biological replicates. The relative expression level was calculated while using the 2−∆∆CT method (Livak & Schmittgen, 2001). The actin and CAC genes were used as internal control (Obrero et al., 2011).

Results

Identification and sequence analysis of VQ genes in C. pepo

After manually removing redundant entries through screening (manually delete the genes without VQ motif) and validation of the search results, a total of 44 VQ genes were identified within the whole genome of C. pepo, which were named CpVQ1–CpVQ44 based on their physical locations on the chromosomes. The characteristics concerning each gene, including gene number, length of coding sequence and amino acid sequence, MW and PI of the proteins, are summarized in Table 1. Subsequent sequence analysis of these 44 CpVQs showed that the encoded CpVQ proteins ranged from 73 amino acids (aa) (CpVQ34) to 628 amino acids (CpVQ14) in length (average length: 219 amino acids). Similar to previous studies in Arabidopsis and O. sativa, most VQ proteins contained less than 300 amino acids. The calculated molecular weight and isoelectric points of these proteins varied from 8.34 kDa (CpVQ34) to 67.83 kDa (CpVQ14), and 4.68 (CpVQ40) to 10.67 (CpVQ19), respectively. The length of the coding sequences of this gene family was between 222 and 1,887 bp (average length: 660 bp).

Table 1 List of all VQ genes identified in C. pepo.

Gene name	Gene locus	Chromosome location	Length (aa)	pI	Molecular weight (Da)	Family group	
CpVQ1	Cp4.1LG00g04520	16693604–16694523	160	5.44	17,313.57	III	
CpVQ2	Cp4.1LG00g15310	42412302–42412658	118	9.83	13,171.89	II	
CpVQ3	Cp4.1LG01g01160	LG01: 3157885–3158544	219	9.03	22,938.39	V	
CpVQ4	Cp4.1LG01g09330	LG01: 4248892–4250868	389	6.8	41,553.79	V	
CpVQ5	Cp4.1LG01g21070	LG01: 17809037–17809507	156	8.13	17,422.62	II	
CpVQ6	Cp4.1LG02g02700	LG02: 3577192–3577794	200	6.91	21,690.73	V	
CpVQ7	Cp4.1LG02g03510	LG02: 3102280–3103518	283	9.16	30,711.29	V	
CpVQ8	Cp4.1LG02g05130	LG02: 2005092–2009226	118	9.83	13,178.19	II	
CpVQ9	Cp4.1LG03g04140	LG03: 2346972–2347676	234	6.53	25,197.91	VI	
CpVQ10	Cp4.1LG04g14070	LG04: 11310784–11311509	241	5.94	25,762.45	II	
CpVQ11	Cp4.1LG05g10550	LG05: 7053658–7054299	213	6.51	22,984.1	II	
CpVQ12	Cp4.1LG06g02740	LG06: 1556739–1557260	173	10.13	18,950.64	IV	
CpVQ13	Cp4.1LG06g04260	LG06: 2472160–2473164	334	9.91	35,686.43	VII	
CpVQ14	Cp4.1LG06g05110	LG06: 3011929–3017926	628	5.57	67,834.1	V	
CpVQ15	Cp4.1LG06g05610	LG06: 3325430–3326068	113	9.62	12,485.03	V	
CpVQ16	Cp4.1LG06g06820	LG06: 4263922–4264659	245	8.71	26,210.14	IV	
CpVQ17	Cp4.1LG07g01470	LG07: 802433–803374	313	10.08	33,653.95	VII	
CpVQ18	Cp4.1LG07g09800	LG07: 8797116–8797493	125	6.05	14,220.9	III	
CpVQ19	Cp4.1LG08g09380	LG08: 7407091–7407423	110	10.67	12,069.01	II	
CpVQ20	Cp4.1LG08g13390	LG08: 9672254–9672643	129	4.84	14,612.33	III	
CpVQ21	Cp4.1LG08g13490	LG08: 9735143–9735811	222	5.92	23,933.64	VI	
CpVQ22	Cp4.1LG09g08360	LG09: 7719957–7720586	209	9.64	22,441.4	IV	
CpVQ23	Cp4.1LG10g01130	LG10: 3319719–3322117	425	8.71	45,497.21	V	
CpVQ24	Cp4.1LG10g01930	LG10: 2886496–2886996	166	9.23	18,288.05	VIII	
CpVQ25	Cp4.1LG10g04880	LG10: 1115416–1116003	195	9.05	21,186.84	IV	
CpVQ26	Cp4.1LG10g06710	LG10: 166830–167354	174	6.28	18,860.78	VI	
CpVQ27	Cp4.1LG10g11090	LG10: 7510307–7511283	164	5.72	17,629.89	III	
CpVQ28	Cp4.1LG10g12290	LG10: 9255696–9256184	162	10.05	18,042.53	IV	
CpVQ29	Cp4.1LG11g04560	LG11: 2540172–2540996	274	10.43	30,231.7	VII	
CpVQ30	Cp4.1LG12g07320	LG12: 6924419–6924844	141	5.91	15,294.24	IV	
CpVQ31	Cp4.1LG13g04200	LG13: 6294724–6297911	282	9.99	31,079.58	II	
CpVQ32	Cp4.1LG14g01330	LG14: 3722302–3723387	425	8.71	45,497.21	V	
CpVQ33	Cp4.1LG14g03000	LG14: 2622622–2623233	203	9.68	21,226.55	V	
CpVQ34	Cp4.1LG14g03270	LG14: 2365261–2365482	73	9.15	8,344.47	I	
CpVQ35	Cp4.1LG16g00810	LG16: 1567379–1568005	208	6.3	22,405.61	II	
CpVQ36	Cp4.1LG17g08680	LG17: 5075487–5075909	140	7.84	14,816.74	IV	
CpVQ37	Cp4.1LG19g00160	LG19: 110455–110994	179	10.04	19,803.85	IV	
CpVQ38	Cp4.1LG19g02600	LG19: 2163376–2163876	166	4.93	17,768.9	III	
CpVQ39	Cp4.1LG19g02610	LG19: 2160888–2161388	166	4.93	17,768.9	III	
CpVQ40	Cp4.1LG19g06070	LG19: 7655240–7655563	107	4.68	12,049.37	I	
CpVQ41	Cp4.1LG19g07250	LG19: 7111871–7112545	224	9.62	24,493.82	IV	
CpVQ42	Cp4.1LG19g09780	LG19: 5873440–5875876	344	6.26	36,896.06	V	
CpVQ43	Cp4.1LG19g10580	LG19: 5411418–5411909	163	9.66	17,828.64	VIII	
CpVQ44	Cp4.1LG20g04410	LG20: 2519217″2520794	335	8.85	36,902.06	IV	

Phylogenetic analysis of CpVQs

To detect the evolutionary relationships and classification of the VQ family in C. pepo, unrooted phylogenetic Neighbor-Joining trees were constructed with the 44 CpVQ proteins and the known VQ protein from Arabidopsis. Through the analysis of the phylogenetic and structural features of the VQ domains, these proteins were divided into eight clades (I–VIII) based on the nomenclature of the Arabidopsis VQs, with two proteins in I and VIII groups, three each in VI and VII, six members in III, eight in II, and group IV and V have the biggest amount of proteins with 10 VQs (Fig. 1). When compared with the other groups, the size of group IV and V were significantly larger. These results were not completely consistent with previous studies in A. thaliana, O. sativa, and Zea mays. In addition, a phylogenic tree was built using the 44 CpVQ protein sequences, which indicated that the groups IV and V in C. pepo had more VQ members than those in Arabidopsis. In addition, it is also possible that the expansion of the two subgroups may have resulted from gene duplications.

Figure 1 Phylogenetic tree analysis of the VQ genes in C. pepo and Arabidopsis thaliana.

The clusters were designated as group I–VIII and indicated in a specific color.

Gene structure and conserved motifs analysis

Gene structure can provide more information about the evolutionary relationship in a gene family; thus, the organization of exons and introns in CpVQs was investigated using the GSDS2.0 online tool. Structural analysis of VQs showed that half of the members of group V had introns, such as CpVQ14 that had four introns, which may be related to their long sequences; genes in group VII have longer coding regions; whereas genes of group I have shorter coding regions than other groups. Moreover, 75% of CpVQs were found to be intronless genes, suggesting that many introns were lost during the long evolution. To further analyze the characteristics of CpVQs, the MEME online tool was used to predict the potential conserved motifs. A total of 20 motifs were identified, with lengths ranging between nine and 48 amino acids. Noteworthy, every CpVQ contained 1–6 conserved motifs (Fig. 2). Motif 1 contained the domain present in every VQ member; thus, representing an essential element in the VQ family. In addition, every group was found to have a clearly identifiable motif structure that distinguished it from the other groups. For example, Motif 2 and Motif 5 were prominent in Group IV, Motif 17 was only observed in Group VII, and Motif 6 was only present in Group III. It is worth mentioning that the gene structure and motif organization of CpVQs were similar within the same group, but diverged between groups, which also indicated that the phylogenetic classification was reliable.

Figure 2 Phylogenetic tree, conserved motifs and gene structure in CpVQs.

(A) Phylogenetic relationships. (B) Conserved motifs of the CpVQs. Each motif is represented by a number in colored box. (C) Exon/intron structures of CpVQ genes.

Chromosome mapping and duplication events analysis

Two VQ genes (CpVQ1 and CpVQ2) could not be mapped on any chromosome, but the remaining CpVQs were randomly distributed on the chromosomes, except chomosomes 15 and 18 (Fig. 3). Chromosome 19 had the highest number of CpVQs (n = 7), followed by chromosome 10 (n = 6), chromosomes 1, 2, 6, 8, and 14 (n = 3), chromosome 7 (n = 2), and the remaining chromosomes contained one VQ. In addition, most CpVQs were found to be mainly located on the two ends of the chromosomes.

Figure 3 Chromosome location in C. pepo.

Chromosome numbers were indicated above each chromosome. The size of a chromosome was indicated by its relative length.

To better understand the evolution mechanism of VQs in plants, the tandem duplication and segmental duplication events of the CpVQ family were evaluated. Based on BLSTAp results, no tandem duplication event was detected. A total of 21 pairs with 28 CpVQs were identified in the whole genome (Fig. 4). Moreover, groups V and IV contained six and five segmental duplication events, respectively. Hence, these results suggested that segmental duplications were the primary driving force responsible for the expansion of the VQ family in C. pepo, which is consistent with previous reports.

Figure 4 Syntenic analysis of VQ genes.

Gray lines in the background indicate the collinear blocks within the C. pepo and oneself genomes, while the dark grey lines highlight the syntenic VQ gene pairs.

Synteny analysis of VQ genes

To further understand the expansion mechanisms of the CpVQ family, we constructed comparative syntenic maps of C. pepo associated with four representative species, including two dicots (Arabidopsis and Cucumis melo) and two monocots (O. sativa and Z. mays). Overall, the CpVQ genes had the most homologous gene pairs with C. melo (n = 27), followed by Arabidopsis (n = 16). However, no homologous gene pairs were observed between C. pepo and O. sativa or Z. mays (Fig. 5). The syntenic analysis results further suggested that the VQ genes are highly conserved in dicotyledonous plants.

Figure 5 Synteny analysis of VQ genes between C. pepo and plant species.

Synteny analysis of the VQ genes between (A) C. pepo and Arabidopsis thaliana; (B) C. pepo and Cucumis melo; (C) C. pepo and Oryza sativa; (D) C. pepo and Zea mays. Gray lines in the background indicate the collinear blocks within the C. pepo and other plant genomes, while the black lines highlight the syntenic VQ gene pairs.

We calculated the Ka, Ks and Ka/Ks ratios of all para-homologous gene pairs (Table 2) to explore the evolutionary constraints of repeated CpVQs. The Ka/Ks values of most gene pairs were less than 1.0, which indicated that these gene pairs had undergone purification selection pressure. In addition, 21 pairs of genes varied from 31 to 216 MYA.

Table 2 Ka, Ks and Ka/Ks values calculated for homologous VQ gene pairs.

Gene1	Gene2	Ka	Ks	Ka/Ks ratio	Differentiation time	
CpVQ3	CpVQ33	0.143408874	0.41714233	0.343788832	34.19199426	
CpVQ4	CpVQ23	0.56179148	1.733083058	0.324157274	142.0559883	
CpVQ4	CpVQ32	0.114789648	0.572645983	0.200454821	46.93819533	
CpVQ5	CpVQ31	0.244725293	0.735263756	0.332840142	60.26752102	
CpVQ6	CpVQ15	1.018237411	1.77385189	0.574026172	145.3976959	
CpVQ8	CpVQ13	0.074887932	0.430075478	0.174127416	35.25208839	
CpVQ8	CpVQ17	0.302766804	1.921171796	0.157594862	157.473098	
CpVQ8	CpVQ29	0.319999352	1.905949455	0.167894983	156.2253652	
CpVQ9	CpVQ21	0.895462518	NaN	NaN	#VALUE!	
CpVQ11	CpVQ35	0.083226176	0.468993438	0.177457016	38.4420851	
CpVQ13	CpVQ17	0.304326774	1.957821306	0.155441548	160.4771563	
CpVQ13	CpVQ29	0.301640689	2.424748395	0.12440082	198.7498684	
CpVQ16	CpVQ22	0.351584842	2.560617395	0.137304715	209.8866717	
CpVQ22	CpVQ25	0.193161647	1.846190406	0.104627153	151.3270824	
CpVQ22	CpVQ41	0.206138134	1.465932276	0.140619139	120.1583833	
CpVQ23	CpVQ32	0.614828236	2.638504545	0.233021481	216.2708644	
CpVQ23	CpVQ42	0.202547017	0.536664673	0.377418203	43.98890764	
CpVQ24	CpVQ43	0.142828118	0.572200927	0.249611826	46.90171536	
CpVQ25	CpVQ41	0.111174117	0.440146678	0.252584246	36.07759653	
CpVQ27	CpVQ39	0.182897122	0.455285224	0.401719872	37.31846099	
CpVQ30	CpVQ36	0.207558725	0.385917729	0.537831536	31.6326007	

Expression pattern of the CpVQ genes in different tissues

In order to explore the possible role of CpVQ gene in the growth and development of C. pepo, we performed qRT-PCR expression analysis in three tissues of roots, stems and leaves. Expression patterns varied among the randomly selected 15 CpVQ genes (Fig. 6). Five CpVQ genes (CaVQ1, 3, 12, 21 and 34) were highly expressed in the root; two CpVQ genes (CpVQ9 and 22) were highly expressed in leaf and root; one CpVQ genes CpVQ40 were highly expressed in stem and root; three CpVQ genes (CpVQ16, 26 and 39) were highly expressed in stem.

Figure 6 (A–O) Expression analysis of the CpVQ genes in different tissues of C. pepo.

The surveyed tissues include root, stem, leaf. The 2−ΔΔCt method was used to calculate the expression levels of target genes in different tissues.

Expression pattern of the CpVQs under powdery mildew stress

Several studies have shown that VQs play an important role in plant response to abiotic and biotic stress. Therefore, the expression profiles of CpVQs under powdery mildew stress were evaluated next based on RNA-seq data. Most of the CpVQs were found to be downregulated at early treatment timepoints, such as the members in group I and IV (Fig. 7). In addition, some members were downregulated and then upregulated. A total of 15 CpVQs from different groups were randomly selected for further validation by qRT-PCR (Fig. 8). Accordingly, most of the tested CpVQs were confirmed to be downregulated after powdery mildew infection, with the expression of these genes being correlated with the RNA-seq data at different treatment timepoints. Among these genes, the expression of two genes (CpVQ1, CpVQ39) significantly increased (more than 2-fold) at 12 h, whereas the expression of eight genes (CpVQ3, CpVQ9, CpVQ12, CpVQ16, CpVQ21, CpVQ22, CpVQ36, and CpVQ40) was significantly downregulated. In addition, three genes (CpVQ26, CpVQ33, and CpVQ34) had a high expression value at 24 h after powdery mildew stress.

Figure 7 Expression profiles of CpVQs under powdery mildew stress based on the RNA-seq data.

The gene expression values are square-root transformed fragments per kilo-bases per million mapped reads (FPKM). Different colors in map represent gene transcript abundance values as shown in the color bar.

Figure 8 qRT-PCR validation of VQ genes in the response to powdery mildew treatment.

Stress treatments and time course are described in “Materials & Methods”. (A–O) Different genes that were evaluated by qRT-PCR. Asterisks indicate statistically significant differences between the stressed samples and counterpart controls (*p < 0.05, **p < 0.01).

CpVQs gene expression following abiotic stress by qRT-PCR

To further investigate the role of CpVQs in abiotic stress responses, We randomly selected 15 CpVQ genes from eight groups, and made sure their responses to the drought-, cold-, salt-, and waterlogging-stress.

Under drought treatment (Fig. 9), all CpVQs were upregulated at different treatment timepoints. The expression of six genes (CpVQ1, CpVQ14, CpVQ26, CpVQ33, CpVQ34, and CpVQ40) significantly increased (more than 5-fold) at 12 h, whereas CpVQ21 was significantly downregulated at 12 h but it then increased at 24 h. In addition, five genes (CpVQ3, CpVQ8, CpVQ9, CpVQ16, and CpVQ39) had a high expression value at 24 h after drought stress.

Figure 9 qRT-PCR validation of VQ genes in the response to drought treatment.

Stress treatments and time course are described in “Materials & Methods”. (A–O) Different genes that were evaluated by qRT-PCR. Asterisks indicate statistically significant differences between the stressed samples and counterpart controls (*p < 0.05, **p < 0.01).

During cold stress (Fig. 10), the expression of eight CpVQs (CpVQ1, CpVQ16, CpVQ22, CpVQ26, CpVQ33, CpVQ34, CpVQ39, and CpVQ40) were significantly upregulated (more than 2-fold) at 24 h. In contrast, four genes (CpVQ3, CpVQ9, CpVQ21 and CpVQ36) were downregulated at 12 h and their levels were further decreased at 24 h. Three members (CpVQ8, CpVQ14, and CpVQ34) were downregulated at early time points but their expression were significantly increased and peaked at 24 h.

Figure 10 qRT-PCR validation of VQ genes in the response to cold treatment.

Stress treatments and time course are described in “Materials & Methods”. (A–O) Different genes that were evaluated by qRT-PCR. Asterisks indicate statistically significant differences between the stressed samples and counterpart controls (*p < 0.05, **p < 0.01).

Upon salt stress (Fig. 11), six genes (CpVQ16, CpVQ22, CpVQ26, CpVQ33, CpVQ34, and CpVQ40) shared a similar expression trend, rapidly rising at early time points but with subsequently decreased expression. In contrast, CpVQ8 and CpVQ12 showed a trend of downregulation from 0 to 12 h, but then their levels were rapidly increased. In addition, the expression of CpVQ21 showed a trend of downregulation from 0 to 24 h.

Figure 11 qRT-PCR validation of VQ genes in the response to salt treatment.

Stress treatments and time course are described in “Materials & Methods”. (A–O) Different genes that were evaluated by qRT-PCR. Asterisks indicate statistically significant differences between the stressed samples and counterpart controls (*p < 0.05, **p < 0.01).

For the case of waterlogging treatment (Fig. 12). The four genes (CpVQ3, CpVQ9, CpVQ21 and CpVQ26) upregulated in the first 12 h and then downregulated in the next 24 h. In contrast, CpVQ8, CpVQ12 and CpVQ14 downregulated in the first 12 h and then upregulated in the next 24 h. Interesting, the expression of seven CpVQs (CpVQ1, CpVQ16, CpVQ22, CpVQ33, CpVQ34, CpVQ39, and CpVQ40) were upregulated from 0 to 24 h.

Figure 12 qRT-PCR validation of VQ genes in the response to waterlogging treatment.

Stress treatments and time course are described in “Materials & Methods”. (A–O) Different genes that were evaluated by qRT-PCR. Asterisks indicate statistically significant differences between the stressed samples and counterpart controls (*p < 0.05, **p < 0.01).

Discussion

VQ has been proven to be the main transcriptional regulator in plants. Currently, the VQ family genes have been systematically analyzed in Arabidopsis (Cheng et al., 2012), rice (Kim et al., 2013a), maize (Song et al., 2016), tobacco (Liu et al., 2020) and cotton (Chen et al., 2020b), and responding to biotic and abiotic stresses (Jiang, Sevugan & Ramachandran, 2018; Perruc et al., 2004). However, current information on VQ characteristics in C. pepo is limited. Therefore, a comprehensive analysis of VQ genes in C. pepo and their expression patterns under various non-biological and powdery mildew treatments may pave the way for a better understanding of the mechanism of plant growth and development. This will also help to select candidate genes and lay the foundation for further in-depth study of the role of different regulatory networks in plant development and stress-related processes.

Conservation of the VQ gene family of C. pepo

In higher eukaryotes, genes without introns are very common (Louhichi, Fourati & Rebaï, 2011; Liu et al., 2021). Herein, according to the gene structure, most VQ genes in C. pepo were also found to have no introns. Only 11 VQs had introns, among which CpVQ14 contained four introns. This result is consistent with the lack of introns in 88.2% of Arabidopsis, 90% of Chinese Cabbage, and 92.3% of tomato (Cheng et al., 2012; Zhang et al., 2015; Ding et al., 2019). Subsequently, a phylogenetic tree was constructed based on the VQ proteins of C. pepo and A. thaliana. Through the analysis of the phylogenetic and structural features of the VQ domains, and these proteins were divided into eight clades (I–VIII) based on the nomenclature of the A. thaliana VQs. This analysis revealed that VQs with introns are located in different subfamilies, suggesting that these introns appear relatively independently. Comparative these plants (Cicer arietinum and Medicago truncatula, A. thaliana and lower plants, moss) indicate that most VQ genes have lost introns during the long evolutionary period. Taken together, VQs identified in C. pepo and in other species provide a certain reference value for the evolution of introns in plants. The average length of the CpVQ is 219.3 amino acids, a majority of the C. pepo VQ genes are intronless so they encode relatively small proteins with fewer than 300 amino acid residues, which is highly similar to those reported for Nicotiana tabacum, Arabidopsis, tomato, O. sativa, and other plants (Liu et al., 2020; Cheng et al., 2012; Ding et al., 2019; Kim et al., 2013a). Noteworthy, CpVQ4, CpVQ14, CpVQ23, and CpVQ32 were all found to be close to 400 amino acids.

The genetic structure of the VQ gene and the conserved Motif not only appear in higher plants such as Arabidopsis, O. sativa, Z. mays, among others, but also in lower plants such as moss, which also implies their ancient origins in the evolutionary history and their important role in plant development (Jing & Lin, 2015; Song et al., 2016). Moreover, all 44 VQ proteins of C. pepo were found to harbor conserved VQ domains and contained the same type of Motifs, implying that in the same branch of the evolutionary tree, the closer the genes of VQ protein are, the more similar the gene structure is. Motif 1 was identified as the core Motif that composes the VQ domain, which was included in all pumpkin VQ proteins, thereby endowing the pumpkin VQ protein with specific biological functions. These results were in agreement with those reported for tomato and soybean species (Ding et al., 2019; Wang et al., 2019). In addition, VQ proteins with similar Motif composition were also found to be located in the same sub-branch of the evolutionary tree, as the Motifs composition between different branches was different.

Expansion mechanism of the C. pepo VQ family

Genome replication events play an important role in expanding the size of the genome and diversifying gene functions, as replication events can result in genes with new functions (Rensing, 2014). In higher plants, tandem repeat events and chromosome fragment replication events are the main processes contributing to gene family expansion (Wang, Wang & Paterson, 2012b). Previous studies have shown that diverse WGD events lead to the different sizes of plant genomes (Adams & Wendel, 2005). In this study, 44 VQ genes were identified in C. pepo, which is a slightly high number of genes as compared with other species, such as Arabidopsis with 34 VQ members, or O. sativa and Vitis vinifera with 39 and 18 VQs, respectively (Cheng et al., 2012; Kim et al., 2013a; Wang et al., 2015a). Nevertheless, the number of VQ members in C. pepo was lower than initially predicted while considering its genome size of 261 Mb, as compared with the 125, 389, 486 Mb of Arabidopsis, O. sativa, and V. vinifera, respectively. Therefore, it can be concluded that there is no necessary connection between the size of the genome and the number of family members. Previous studies have shown that chromosome fragment replication is considered to be the main mechanism of VQ gene expansion (Tang et al., 2008); thus, the evolutionary process can explain the number of specific VQ genes in a species, not the size of the genome. The main process for the expansion of the C. pepo VQ family was found to be chromosome fragment duplication events, which is consistent with the results of previous studies. Among the 44 CpVQs, a total of 28 members participated in 21 chromosome fragment duplication events, accounting for 63.6%, and no tandem duplication event was identified. This is consistent with the expansion of the VQ family in Brassica napus, which is driven by chromosome fragment duplication, with tandem repeat events as the second driving force (Zou et al., 2021). Gene duplication can produce gene function redundancy, and these repeated genes can develop different gene expression patterns. We calculated the Ka, Ks and Ka/Ks ratios of all para-homologous gene pairs to explore the evolutionary constraints of repeated CpVQs. The Ka/Ks values of most gene pairs were less than 1.0, which indicated that these gene pairs had undergone purification selection pressure. In the present study, some synlinear genes showed different expression patterns, such as CpVQ3 and CpVQ33, CpVQ9 and CpVQ21, including under drought treatment, suggesting that these homologous genes in C. pepo may have different functions in regulation the normal growth and development of plants.

Expression patterns of VQ members of C. pepo

Previous studies have shown that members of the VQ family play an important role in the entire plant development process and respond to various biotic and abiotic stresses. Molecular genetic evidence suggests that the plant VQ protein may be an important regulator of disease resistance and tolerance (Wang et al., 2015a). In this study, we analyzed the expression level of CpVQ gene in different tissues of C. pepo (Fig. 6). The results showed that most genes were differentially expressed in the tissues we analyzed. It indicates that the CpVQ gene may play an important role in the growth and development of these organs or tissues. In addition, based on transcriptome data, the expression VQs in C. pepo leaves under powdery mildew treatments at different timepoints was evaluated (as observed in Fig. 7). Most VQ members showed a significant decline in expression after 12 h of powdery mildew treatment, however, the expression patterns of the third group is different from the other groups. In the third group, except CpVQ20, all were significantly up-regulated upon powdery mildew infection, and three genes (CpVQ18, 27 and 38) had higher expression at 24 h, which may be because these genes responded late to powdery mildew stress. Studies have shown that SIB1(AtVQ23) and SIB2(AtVQ16) function as activators in plant defense against necrotrophic pathogens (Lai et al., 2011), so the CpVQ genes classified in the third group may have the same function. In contrast, after 24 h of powdery mildew treatment, the expression of most members in other groups decreased significantly. Similar results have been reported in grapevine that VvVQ genes are quickly responsive to powdery mildew stresses (Wang et al., 2015a). Therefore, the members of the VQ family in C. pepo may play a key role in the powdery mildew signaling pathway. Moreover, analysis of VQs expression under different abiotic stresses further demonstrated that the members of the VQ family are also significantly induced under different stresses. In this study, four CpVQ genes (CpVQ3, 9, 16, and 22) significantly upregulated during drought stress (as observed in Fig. 9), which results are similar to the OsVQ genes that 22 OsVQ genes are upregulated under drought stress (Kim et al., 2013a). Under cold stress (as observed in Fig. 10), five CaVQ genes (CaVQ1, 22, 33, 39 and 40) were significantly induced. Similar results have been reported in Chinese cabbage that BrVQ genes are quickly responsive to cold stresses (Zhang et al., 2015). In addition, the VQ genes are also sensitive to salt changes. Two CpVQ genes (CaVQ3 and 9) were upregulated during salt treatment (as observed in Fig. 11). Similar changes occurred in Arabidopsis. AtVQ9 and AtVQ15 were significantly expressed under salt stress. Seven CaVQ genes (CaVQ1, 16, 22, 33, 34, 39 and 40) were upregulated at waterlogging treatment (as observed in Fig. 12). In summary, CpVQ members may be involved in regulating the response of plants to various abiotic stresses and powdery mildew stress, and their response mechanisms may be complex and diverse.

In summary, this study has systematically analyzed the evolutionary relationship, conserved structure, and expression patterns of the members of the VQ family of C. pepo at the whole genome level. The selection of candidate genes can provide reference for future investigations.

Conclusions

In conclusion, this study provides the first comprehensive and systematic analysis of the 44 VQ genes identified in C. pepo genome. All CpVQ genes can be divided into eight groups (I–VIII), and were found to expand by chromosome fragment duplication events. RNA-seq analysis further showed that half of the VQs have significantly different expression patterns at different timepoints of powdery mildew infection. Therefore, the members of the VQ family in C. pepo may play a key role in the powdery mildew signaling pathway. Genetic analysis data further confirmed that the VQ family members respond to different abiotic stresses. Taken together, these findings provide a theoretical basis for further research on the functions of CpVQs.

Supplemental Information

Supplemental Information 1 Raw data.

Click here for additional data file.

Supplemental Information 2 List of primers used in qRT-PCR.

Click here for additional data file.

Additional Information and Declarations

Competing Interests

Author Contributions

Data Availability

The authors declare that they have no competing interests.

Ke Xu conceived and designed the experiments, performed the experiments, analyzed the data, prepared figures and/or tables, authored or reviewed drafts of the paper, and approved the final draft.

Ping Wang conceived and designed the experiments, authored or reviewed drafts of the paper, and approved the final draft.

The following information was supplied regarding data availability:

The raw measurements are available in the Supplemental File.

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
