# Peer review of "Genome-wide identification and expression analysis of the VQ gene family in Cucurbita pepo L"

_PeerJ, doi:10.7717/peerj.12827_

## Round 0.1 · original submission · Major Revisions

Please address the concerns of all reviewers and revise the manuscript accordingly.

Reviewer 1 ·

Basic reporting

The use of English language, grammar and phrasing of sentences is quite satisfactory. The paper was easy to comprehend and had a clear flow to it. The authors do a good job of introducing the properties of C. pepo L that make it valuable as well as the importance of studying the VQ gene in plant stress regulation.

Experimental design

The study was conducted quite thoroughly especially in documenting the gene expression changes and molecular responses against various stress factors like drought, powdery mildew, cold, salt on Cucurbita pepo L. The data seems backed by various statistical analysis which makes the findings realistic and reliable.

The authors show extensive use of bioinformatics tools and techniques as well as provide specific versions of each tool used which would help in reproducibility of the results.

Validity of the findings

The study seems to be a first of its kind to throw light on VQ and its role in protecting Cucurbita pepo L under abiotic stresses. What is actually interesting to note that the unsupervised clustering of CpVQs under powdery mildew stress is able to classify each of the genes into their specific phylogenetically classified group.

Reviewer 2 ·

Basic reporting

Authors should provide relevant references they are missing throughout the manuscript. Please provide relevant references for the statements, one such example is Line 37-40. Another example is Line 45-47 "In Arabidopsis… plants".

Line 70-71 "widely .. Worldwide", "in" should be dropped. Authors should do a thorough proof-reading of the manuscript and use unambiguous English language for the international readers to follow.

Line 132 Title "VQ gene…" formatting is not consistent, there are several instances where the manuscript formatting is not up to the journal standards. Please use consistent formatting across the entire manuscript.

Experimental design

Methods described in the manuscript are relatively clear, but authors should address the below concerns for improvements.

Line 92 authors should provide full forms of Abbreviations Example: "CDD"

Line 99 "multiple sequence aliment" should be multiple sequence alignment, please correct similar mistakes across the manuscript.

Line 103-104 “The phylogenetic tree was edit and illustrate by iTOL (http://itol.embl.de/help.cgi) online tool”. Please correct the English language and rephrase.

Validity of the findings

Lines 84-92, “HMM profiles of the VQ motif (PF05678)…. (E-value) cut-off was set as 0.1 (Punta et. al, 2012). The Arabidopsis Information Resource (TAIR) database (http://arabidopsis.org) was used as queries to conduct a local BLASTP against the protein database of Cucurbita peop. The targeting genes with similarity of E-value less than 1e-20 were retained for the further analysis". Authors, please elaborate on why different e-values were used?

Line 167-168 Authors should explain what do they mean by "After manually removing redundant entries through screening and validation of the search results". What screening criteria and validation were done before deleting certain entries?

Line 183 Authors mentioned VQ proteins were divided into 8 groups, they should elaborate on how these 8 groups were identified?

Line 281-288 "Construction …… conditions" This result section is not clearly written, and the description of Figure 12 is vague. Currently in my opinion it is not adding any value to this manuscript. Please rewrite and provide context for the readers.

Additional comments

This study investigated the VQ genes in the C. pepo L. genome and analyzed their phylogeny. In addition to this expression level of CpVQs was analyzed under abiotic and biotic stresses. The manuscript is not easy to follow. Please thoroughly proof-read the manuscript and use unambiguous and professional English. Overall, the manuscript requires major revision and clarifications before acceptance. I have included my comments in the various section along with some general comments below.

1) Introduction needs improvement and should be rewritten to provide sufficient background to determine the significance of this work. Currently, it's hard for the reader to follow as there is no connection or flow from one statement to another.

2) Similar research work is conducted on VQ genes in 2016,2019( https://doi.org/10.3389/fpls.2015.01177, Doi: 10.7717/peerj.7509). Authors should comment and discuss how this manuscript adds to what is already known in the field.

3) Genome-wide identification and expression analysis of the VQ gene family have been carried out in many plant species as referenced above. The authors should make relevant comparisons among published VQs and their identified VQs.

4) Authors should discuss the difference in expression trend for VQ Group 3 compared to all other groups (as observed in Fig. 6)? Why group 3 has upregulated expression later in time after powdery mildew stress?

5) Line 297-299 Authors should clarify what they mean by "deeper analysis and investigations" What kind of analysis will this study support? Please provide specific details.

6) Line 355-363 "In this study …..Therefore, the members of the VQ family in C. pepo L. play a key role in the powdery mildew signaling pathway." It is not evident from this paragraph how the authors came to this conclusion? Authors should reframe or provide additional evidence supporting this statement.

·

Basic reporting

The manuscript is very poorly written. Some of the details is given below.

Experimental design

The authors have not mentioned the nature of replicates.
Actin for RT control has been used from other plant sps.
Detail is given below in additional comments.

Validity of the findings

Biological replicates should be added in RT PCR validation.
Incomplete short sequences should be completed by amplification and sequencing or TSA/EST/Nr blast, if available in the database.

Additional comments

The Ms entitled ‘Genome-wide identification and expression analysis of the VQ gene family in Cucurbita pepo L.’ is very poorly written. It should be revised thoroughly for grammatical and technical errors..
1. Correct the name of crop as ‘Cucurbita pepo’ in the entire manuscript. It is written C. peop at many places.
2. Write down the botanical name of plants in the entire manuscript and italicize them.
3. Articles missing at many places. Include ‘The’ before the name of Kits and Tools used in the Ms.
4. When CpVQs are written in context to protein, they should not be italic. Only gene name will be italicized.
5. Lot of grammatical errors have been done in the manuscript. So, here is a need of deep revision of every line.
6. Firstly, write down the full forms with abbreviation in bracket like molecular weight (MW), then only abbreviation should be used in the entire manuscript. Same can be used for other words.
7. Figure 2 and 3 are not clearly visible.
8. The results and discussion part should be elaborated.
Abstract:
1. Line 17 should be rewritten.
Introduction:
1. Lines 64 to 65 found confusing. It should be rephrased.
Materials and Methods:
1. In Line no. 109, CpVQs should not be italicized.
2. The reference cited in line no. 131 not found to be relevant with respective line.
3. Treatment was in biological or technical replicates, explain properly.
4. TBTOLLS citation is missing. Even link is not given.
5. Why C. moschata actin is used as internal control? Why not from the same plant gene?
6. I could not see any statistical analysis for RT data.
Results:
1. In line no. 170, the gene length and length of coding sequence are the characteristics of gene, not protein. It should be corrected.
2. Why there is much variation in the length of proteins? It might be incomplete sequence. Try to complete the length by TSA blast or amplification and sequencing of short genes.
3. In line no.180, the VQ should not be italicized as here it represents the family name, not gene.
4. In line no. 182, VQ genes should be replaced with VQ proteins.
5. The line no. 189-190, don’t found to be relevant. It should be rephrased.
6. Line no. 193-194, the function of gene cannot be determined from gene structure studies (number of exons and introns). So, this line should be corrected.
7. Line no. 207-208, does not make any sense. So, it should be rephrased.
8. In line 211, the heading should be replaced as “chromosome mapping and duplication events analysis”.
9. Line 212 doesn’t find to be relevant. Therefore, it should be replaced.
10. Mention the gene distributions on chromosome 6.
11. Line 220, “BLSTAP” should be replaced with “BLASTp”.
12. Line 237 is confusing. It should be rewritten.
13. The results regarding “Expression pattern of the CpVQs under powdery mildew stress” were concisely written. Therefore, these results should be elaborated.
14. Line 254-255 should be corrected because CpVQ36 gene exhibit upregulation at 0 h of stress treatment, which indicate that it might be involve in early response.
15. Line no. 255 and 256, should be corrected, because the genes CpVQ1, CpVQ12, CpVQ14, CpVQ26, CpVQ33, CpVQ34, and CpVQ40 are not significantly increased as displayed in figure 8.
16. Line no. 260-261 should be corrected. The reported genes in your manuscript are not found to be significantly upregulated at early hours of stress as shown in the Figure 9.
17. Line no. 271-272, should be mentioned clearly.
18. Line no.282-283 should be corrected.
19. The results of expression analysis have been written concisely. Therefore, the results should be elaborated and written with more clarity.
Discussion:
1. Discussion should be elaborated. Most of results has not been discussed.
2. Author should discuss the number of VQ genes in other plants also, so that the comparative idea will be available for their distribution in various plants. Why higher number of VQ genes in C. pepo than other dicot plants? What is relation of ploidy level and genome size with the number of VQ gene in various plants?
3. The line no.309-310 should be rewritten. Discuss the phylogenetic relationships with more points and in clear manner.
4. Line no. 319-321 is not clear. It should be rephrased.
5. Line 368-369 should be grammatically corrected.

Figure quality is very poor. Some figures are not visible properly such as figures 2,3 7-11, etc.
Figure legends should also be elaborated.
Why expression analysis is done in only stress treatment not in any tissue developmental stages?

---

## Round 0.2 · Major Revisions

Both reviewers are not satisfied with your revision. Please pay very close attention to reviewer #2, as they raised serious concerns about your controls. Furthermore, the reviewers have identified that the English language must be improved.

Reviewer 2 ·

Basic reporting

There are still some minor grammatical errors in the manuscripts that need to be addressed.

Experimental design

Experimental design is good enough after modification done by authors as pointed in my previous report.

Validity of the findings

Additional clarifications as requested by reviewers are now supplied and Irrelevant information such as Fig12 is removed. The introduction is sufficiently clear after being re-written.

Additional comments

Overall, the authors have satisfactorily addressed all of the reviewer’s concerns.

·

Basic reporting

The Ms still needs improvement.

Experimental design

In RT-PCR, earlier they have written actin gene as internal control, now they are writing CAC gene. Either it was negligence of author, or they have wrongly mentioned about the control gene.
What is this CAC gene? why it has been used as control? is there any evidence of homeotic expression of this gene? Is it a gold standard for expression analysis? I don't think so. In that case, authors should use more than one internal control, which has been regular practice now a day.

Validity of the findings

Authors did not seems to be consistent at there own reports.

Additional comments

Manuscript is still very poorly written. Even, they have still not correctly written the name of his studies plant at all the places. For instance at line 213, C.peop is written in place of C. pepo. If they can not write the correct name of the studied plant, then how one can expect better Ms from them.
Even in title, despite of suggestion to italicize the gene name, it is still not done. Is VQ not the gene name in title?
Authors have highlighted all the Ms, in place of highlighting the changes, how one can see the actual changes from the earlier Ms?

---

## Round 0.3 · accepted · Accept

All critiques were addressed and revised manuscript is acceptable now.

Reviewer 2 ·

Basic reporting

Grammatical errors were addressed

Experimental design

Experimental design is good enough after modification done by authors as pointed in my previous report.

Validity of the findings

Additional clarifications as requested by reviewers are now supplied